# Effects of Dairy Lambs’ Rearing System and Slaughter Age on Consumer Liking of Lamb Meat and Its Association with Lipid Content and Composition

**DOI:** 10.3390/foods11152350

**Published:** 2022-08-05

**Authors:** Enrique Pavan, Susan A. McCoard, Michael Agnew, Renyu Zhang, Kevin Taukiri, Mustafa M. Farouk, Carolina E. Realini

**Affiliations:** 1AgResearch Limited, Te Ohu Rangahau Kai, Massey University Campus, University Ave., Palmerston North 4474, New Zealand; 2Unidad Integrada Balcarce (Estación Experimental Agropecuaria Balcarce, Instituto Nacional de Tecnología Agropecuaria—Facultad de Ciencias Agrarias, Universidad Nacional de Mar del Plata), CC 276, Balcarce 7620, Argentina; 3AgResearch Limited, Grasslands Research Centre, Private Bag 11008, Palmerston North 4442, New Zealand; 4AgResearch Limited, Ruakura Research Centre, 10 Bisley Road, Hamilton 3214, New Zealand

**Keywords:** rearing system, slaughter age, fatty acids, overall liking, flavor liking, liking of tenderness

## Abstract

The effects of the rearing system (artificially vs. naturally milk-fed) and the slaughter age (3-weeks milk-fed vs. 3-months pasture-fed) on consumer liking of East-Friesian-cross dairy lamb *Longissimus lumborum* muscle and its association with lipid content and composition were evaluated. The artificially reared lambs were removed from their dams at 2–3 days of age and reared with cow milk. Intramuscular fat content (2.8%) was similar between treatments. Only 3 of the 25 fatty acids evaluated were influenced by the rearing system and 15 by the slaughter age. The rearing system had a minor impact (*p* < 0.10), but the slaughter age had a major (*p* < 0.01) impact on consumer liking. All consumers preferred on average meat from 3-weeks-old lambs. However, based on overall liking scores, Cluster-1 (60% of consumers) preferred meat from 3-weeks-old lambs driven by all sensory attributes but mostly tenderness, whereas Cluster-2 preferred meat from 3-months-old lambs driven by flavor only, indicating a preference for stronger flavor from older lambs finished on pasture. Meat fatty acid profile and consumer liking were not influenced by the rearing system but by lamb slaughter age, showing a niche product opportunity for the 3-weeks milk-fed lambs.

## 1. Introduction

New Zealand is well recognized for its lamb meat production and export, but its dairy sheep industry is fairly new [1]. Meat lambs are commonly weaned in New Zealand at 3 to 4 months old and slaughtered within 3 to 8 months of age. In contrast to traditional meat sheep systems, lambs from dairy sheep farming systems are either naturally or artificially reared, and weaned off milk from 4 to 5 weeks of age [2]. In general, New Zealand dairy lambs are also slaughtered at 3 to 8 months of age, but there is an emerging market for milk-fed lambs (1–2 months old).

The rearing system of lambs may influence intramuscular fat (IMF) content and its fatty acid composition and, hence, meat sensory characteristics [3,4,5]. Intramuscular fat content can influence meat tenderness and juiciness [6], while fatty acid composition can mainly influence meat flavor [7,8]. Meat fatty acid composition from milk-fed lambs is mainly influenced by the fatty acid composition of their diet, ewe milk or milk replacer [9,10]. However, the residual effect of a rearing system on meat fat content and its composition from dairy lambs finished on pasture, and how these potential changes affect the eating quality of meat, are not known.

It has been suggested in general that consumers prefer meat products that they are more familiar with [11,12]. For example, it was observed that the sensory preferences of European consumers were associated with the main production systems in their countries [11]; consumers from Mediterranean countries preferred milk- or concentrate-fed lambs, whereas consumers from Northern countries preferred pasture-fed lambs. As New Zealand meat production systems (beef and lamb) are mainly pasture based [13], it would be expected that New Zealand consumers would prefer meat from pasture-fed lambs. However, there is no information about consumer preferences for meat from milk-fed lambs compared with meat from lambs finished on pasture in dairy sheep production systems.

Therefore, the objectives of the present study were to evaluate the effects of the lamb rearing system (naturally vs. artificially milk-fed) and slaughter age (3-weeks milk-fed vs. 3-months pasture-fed) on New Zealand consumers’ liking of lamb meat, and its relationship with total and individual meat fatty acid content. It was hypothesized that both effects, rearing system and slaughter age, would have an impact on the fatty acid composition and consumer liking of lamb meat.

## 2. Materials and Methods

This study was conducted at a commercial dairy sheep farm near Taupo, New Zealand. All animal manipulations in this study were carried out in compliance with the institutional Code of Ethical Conduct for the Use of Animals in Research, Testing, and Teaching, as prescribed in the Animal Welfare Act of 1999 and its amendments (New Zealand). The AgResearch Grasslands Animal Ethics Committee approved the manipulations (Approval number 15056).

### 2.1. Animals, Treatments, and Husbandry

This study used a subset of lambs (*n* = 48) reared from a larger cohort study (*n* = 96). A randomized experimental design with 96 twin-born East-Friesian-cross male dairy sheep lambs, sourced from a commercial dairy sheep farm, was used. The lambs were born indoors in a commercial lambing unit and were reared by their mixed-age dams as a twin pair until 2–3 days of age to enable maternal colostrum intake and thus reduce the risk of failure of passive transfer of immunity. Date of birth for all lambs was recorded to enable lambs to be enrolled in the study at the same age over a 10-day period. Lambs were randomly allocated to 1 of 2 treatment groups balanced for date of birth: natural rearing on their dam in an outdoor pasture grazing system (*n* = 48 lambs and their dams; NR group), or artificial rearing (*n* = 48; AR). For the NR group, the lambs were weighed and moved from the indoor lambing shed into one of 3 ryegrass/white clover mixed sward pasture paddocks (>150 kg/DM/Ha) with similar numbers of animals per paddock. The ewes and lambs remained in these paddocks until weaning at 6 weeks. For the AR group, lambs were removed from their dams and transferred to one of 4 indoor lamb rearing pens (*n* = 12/pen) bedded with straw. The lambs were offered a 100% bovine milk protein and fat milk replacer (AnLamb, NZAgBiz, Auckland, NZ; 25.9% protein, 27.6% fat, 39.3% lactose, 21.3 MJ/kg of DM metabolizable energy) mixed warm at a final concentration of 200 g/L and dispensed via an automatic feeder (DeLaval LKF1200, DeLaval, Hamilton, New Zealand) and fed on-demand (i.e., ad libitum). The fatty acid composition of the milk replacer fed to the AR lambs (AnLambTM) was ≤C12:0, 14.5%; C14:0, 12.2%; C14:1, 0.9%; C16:0, 30.2%; C16:1, 1.3%; C18:0, 9.7%, C18:1, 17.3%; C18:2, 0.8%; C18:3, 0.6%, C18:1 trans-11, 2.9; C18:2 cis-9 trans-11, 1.1%; other fatty acids, 6.9%, and unknown fatty acids, 3.9%. Fresh water was available ad libitum for both groups.

At an average of 3 weeks of age, 12 animals per treatment group were randomly selected from each AR pen (*n* = 4) or NR paddock (*n* = 2) and sent for slaughter (3-weeks milk-fed). Only two paddocks of NR could be included in the 3-weeks milk-fed slaughter because lambs in one of the NR paddocks did not reach the minimum weight criteria for slaughter. Solid feeds were then introduced for the remaining lambs (ryegrass hay and grain-based concentrate; Nourish, Ruminate, New Zealand) offered via feeders with sufficient head space for all lambs indoors (AR), as well as outdoors (NR). For the NR group, the ewes were prevented from having access to the lamb meal feeders. The remaining lambs in the AR group, and twin pairs from the NR group, were reared until ~6 weeks of age and then weaned off milk (i.e., automatic feeders or their dams) on the same day and managed thereafter on a ryegrass/white clover mixed sward pasture (1500–3000 kg/DM/Ha). The grain-based concentrate was capped at the ad libitum feeding level recorded in week 2 post-weaning (~280 g/head/day) for a further 2 weeks, then reduced by 25% in week 5 and 50% in week 6 post-milk-weaning. All grain-based concentrate was removed thereafter (i.e., from 12 weeks of age). Lambs from both groups were managed as a single mob from milk weaning until slaughter at around 3 months of age (3-months pasture-fed). The fatty acid composition of the pasture grazed by the NR and AR lambs from weaning until slaughter at 3 months of age was C14:0, 2%; C16:0, 23%; C16:1, 3%; C18:0, 2%, C18:1, 3%; C18:2, 12%; and C18:3, 56%.

All lambs were vaccinated for scabby mouth (Phenax^®^, Virbac, Hamilton, New Zealand) upon trial entry, and against clostridial infections on Weeks 4 and 8 (8:1 Coglavax8, Ceva Animal Health Limited; New Zealand). All lambs in the 3-months pasture-fed group were administered an anthelmintic drench (Boss Hi Mineral, Alleva Animal Health, New Zealand) post-milk weaning on a monthly basis until slaughter to reduce the risk of parasite infection. All lambs reared to 3 months old were tail docked and castrated on Day 28 using local anesthesia and rubber rings.

The 3-weeks milk-fed and the 3-months pasture-fed lambs were slaughtered at commercial abattoirs in New Zealand. At 24 h postmortem, both loins (M. longissimus lumborum, LL) were collected from each animal (*n* = 48) for total and individual fatty acids determination, and for consumer panel evaluation. A slice of ~2 cm thickness was taken from the cranial end of the LL samples for chemical analysis. The remaining LL portion was vacuum packed, aged at the meat plant for 2 d and then frozen at −20 °C for 5 months until consumer sensory evaluation.

### 2.2. Loin Intramuscular Fat and Fatty Acid Analysis

Prior to fatty acid (FA) composition analysis, samples were freeze-dried (FD 80 Cuddon; Freeze dry Blenheim, New Zealand) and ground (Breville spice/coffee grinder, model BCG200, 220–240 V, ~50 Hz 200 W; Palmerston North, New Zealand). Total and individual FAs content of meat were determined using the Optimized One-Step FA derivatization protocol and the FA methyl ester analysis according to Agnew et al. [14].

### 2.3. Consumer Sensory Evaluation

The project was submitted via the Massey University Human Ethics Committee process for consideration and was deemed low risk and not needing full ethics approval (Ethics Notification Number: 4000024436). Informed consent was obtained from all participants. One hundred and thirty-two consumers were recruited in June 2021 using Qualtrics (Qualtrics, Provo, UT, USA). Participants were preselected according to the following criteria: age (18–65 years old) and consumption profile (consume meat including lamb at least once a fortnight). Twelve sessions were run with 12 consumers per session at the Food Experience and Sensory Testing Lab, Massey University, Palmerston North, New Zealand.

Due to the small size of loins from 3-week-old lambs, both loins from each animal were cooked, while only the left loin from the 3-months pasture-fed lambs was cooked for consumer sensory evaluation. Vacuum-packed LL sections were thawed at 4 °C for 48 h. Loins were sous vide cooked at 58 °C for 1 h. After cooking, the loins were removed from the sous vide bag, pat dried using paper towels, and rested for 4 min. Loins were then grilled until reaching 71 °C of core temperature (monitored using a scanning thermometer, Digi-Sense, Cole Palmer, Auckland, New Zealand) in a clamshell hot plate (Roband Aluminum Grill Station 8 Slice Smooth Plates WGSA815S, Nisbets, Auckland) pre-heated at 150 °C. Cooked loins were then cut into six 1 cm-thick slices, and each slice was cut in half obtaining 12 samples (about 1 cm^3^ each) per animal. Meat samples were individually wrapped with aluminum foil pre-coded with 3-digit random codes and kept in a sample warmer at 50 °C (<10 min) until serving. Samples were served monadically in warm (50 °C) ceramic cups. To avoid first order and carryover effects, samples were presented to panelists in different orders according to a Williams square design [15]. The evaluation of the samples was performed in individual sensory booths that had controlled environmental conditions under red light. Consumers were provided unsalted gluten-free crackers and filtered water to cleanse their palate between samples. Each consumer rated liking of flavor, juiciness, tenderness, and overall liking of four lamb samples using a 100 mm non-structured line scale anchored at each end (0: dislike extremely to 100: like extremely).

### 2.4. Statistical Analysis

Meat IMF content and individual fatty acid proportions were evaluated as a split-plot design by ANOVA using Proc Mixed from SAS [16]. The model included the rearing system effect as the main plot, slaughter age effect as the sub-plot, and their interaction as fixed effects. The pen was considered as the error term for the main plot and included in the model as a random effect. The consumer sensory panel scores were evaluated using the same experimental design adding animal and panelist as random effects and IMF as a covariate. Means differences were assessed using Tukey’s multiple comparison test and significance was declared at *p* < 0.05, unless otherwise noted.

To identify the clusters of consumers based on their overall liking scores an agglomerative hierarchical cluster analysis was performed on the square Euclidean distance matrix using XLSTAT 2007 (Addinsoft 2012) software (Addinsoft, Paris, France). The Ward method was used with the center and reduce options by consumers. Consumer demographic and frequency of lamb meat consumption data were summarized for all consumers and by consumer cluster (Table 1) using the FREQ procedure, and differences between clusters were obtained with the Chi-square test using SAS [16]. Consumer panel data were then analyzed, including clusters and their interactions as fixed effects to the above model.

The relationships between the average consumer panel scores per lamb from both clusters, and between the average consumer panel scores and the meat fatty acid profile per lamb, were assessed using the correlation (CORR) procedure in SAS (*n* = 48).

## 3. Results

### 3.1. Intramuscular Fat and Fatty Acid Composition

The IMF content was not influenced by the rearing system nor by the slaughter age or their interaction (*p* > 0.90; 2.75 ± 0.70%; Table 2). No interaction between the rearing system and slaughter age was observed for fatty acid composition (*p* > 0.05). The rearing system had only few effects on fatty acid composition; the C14:1 proportion was 221% higher (*p* < 0.001) in the meat from the AR than from the NR lambs, while C18:1 trans-11 (TVA) and C18:2 cis-9, trans-11 (CLA) proportions were 15% and 22% lower (*p* < 0.05) in meat from AR than from NR lambs, respectively.

The proportions of several fatty acids were influenced by lamb slaughter age. Proportions of total SFAs, C17:0, and C18:0 were increased by 8, 33, and 35%, respectively, in the meat from the 3-months pasture-fed lambs compared with the meat from the 3-weeks milk-fed lambs, whereas the proportion of C14:0 decreased by 51%. Total branched-chain fatty acid (BCFA) proportion was not influenced by slaughter age (*p* = 0.15), but anteiso C15:0 was 25% higher in the meat from the 3-months pasture-fed than from the 3-weeks milk-fed lambs. The total MUFA, C17:1, TVA, and C18:1 cis-9 proportions were 8, 32, 24, and 9% higher in the meat from the 3-months pasture-fed animals, respectively, whereas C14:1 and C18:1 cis-11 proportions were 72 and 28% lower in the meat from the 3-months pasture-fed lambs, respectively. Finally, it was observed that increasing slaughter age reduced by 29% or more the proportion of total and individual PUFAs but did not affect the proportion of C18:3 *n*-3 (2.10% of total fatty acid content). The proportion of total *n*-6 and *n*-3 PUFAs was also reduced (*p* ≤ 0.001) by increasing animal slaughter age. Furthermore, increasing animal slaughter age reduced the ratios associated with Δ-9 desaturation (C16:1/C18:0, C18:1/C18:0, and CLA/TVA). The PUFA/SFA ratios were not influenced by the rearing system but by the slaughter age, whereas the *n*-6/*n*-3 (1.23) ratio was not influenced by the rearing system or the slaughter age. The PUFA/SFA ratio was 61% higher in the meat from 3-weeks milk-fed lambs than from 3-months pasture-fed lambs.

When significant (*p* < 0.05), Pearson’s correlations between IMF content and individual SFAs, BCFAs, or MUFAs were positive, except for C18:1 cis-11, which showed a negative correlation (Table 2). In contrast, Pearson’s correlations between IMF content and total and individual PUFAs were negative (*p* < 0.05), except for CLA, which showed a trend (*p* < 0.10) for a positive correlation.

### 3.2. Consumer Sensory Evaluation

When considering all consumers, the liking of meat tenderness, juiciness, and the overall liking of meat from the 3-weeks milk-fed lambs was 15, 31, and 21% higher (*p* < 0.0001) than from the 3-months pasture-fed ones, with no difference in flavor liking due to slaughter age (*p* = 0.96; Table 3). Additionally, none of the sensory parameters were influenced by the rearing system (*p* > 0.23) or by its interaction with slaughter age (*p* < 0.21).

Two consumer clusters were generated based on their overall liking scores; Cluster-1 comprised 60% and Cluster-2 40% of consumers. There was an interaction between cluster and rearing system for overall liking score (*p* = 0.0126; Table 4); however, when using the Tukey test to separate the means, only a trend (*p* = 0.08) was observed showing that consumers in Cluster-1 preferred meat from NR than from AR lambs, while no difference was observed for Cluster-2 (*p* = 0.64). There was an interaction between cluster and slaughter age for all sensory traits evaluated (*p* < 0.001; Table 4). Cluster-1 had greater overall liking and flavor liking for meat from 3-weeks than from 3-months lambs, while Cluster-2 had a greater preference for meat from 3-months than from 3-weeks lambs. Liking of tenderness and juiciness was greater for meat from 3-weeks than from 3-months lambs in Cluster-1, while no difference between slaughter ages was observed for Cluster-2.

All correlations between the evaluated sensory parameters were highly significant and positive. The overall liking scores had similar correlations with all three liking scores (r = 0.88 to 0.92) in Cluster-1 but showed a higher correlation with flavor (r = 0.93) than with tenderness (r = 0.69) and juiciness liking (r= 0.73) scores in Cluster-2.

### 3.3. Correlations between Consumer Liking Scores and IMF and Fatty Acid Composition of Lamb

Correlations were calculated between consumer liking scores for tenderness, juiciness, flavor and overall liking, and loin IMF content and proportions of fatty acids (data not shown). Consumer liking scores for the parameters evaluated were not correlated with loin IMF content in Cluster-1 (*p* > 0.05), but they were positively correlated (r = 0.37 to 0.45, *p* < 0.05) in Cluster-2. Tenderness liking scores from consumers in Cluster-1 were negatively correlated with the proportion of C18:0 (r = −0.80). Flavor liking scores from consumers in Cluster-1 were positively correlated (*p* < 0.05) with C10:0, C14:0, C18:1 cis-11, CLA, and C22:6 *n*-3 (r = 0.29 to 0.54), and negatively with C17:0, C18:0, C17:1 and C18:3 *n*-3 (r = −0.31 to −0.38). In contrast, flavor liking scores from consumers in Cluster-2 were positively correlated with total SFA and MUFA proportions and with C17:0, C18:0, iso C15:0, C17:1, C18:1 trans-9, TVA, and C18:1 cis-9 proportions (r = 0.30 to 0.59); and negatively correlated with total PUFA proportion and with C18:1 cis-11 and all individual PUFAs proportions, except C18:3 *n*-3 and CLA (r = −0.35 to −0.54).

## 4. Discussion

### 4.1. Rearing System and Slaughter Age Effects on IMF Content

The lack of rearing system by slaughter age interaction and rearing system effects on IMF content suggest the difference in milk composition and intake between the two rearing systems was not sufficient to cause a change in IMF content. In addition, the lack of difference in IMF content between both slaughter ages (3-weeks milk-fed and 3-months pasture-fed) could be associated with the fat mobilization generated by the weaning phase of the 3-months pasture-fed lambs, as suggested by Prache et al. [17] and Ye et al. [18]. Despite the mobilization during weaning, the extra time that the 3-months pasture-fed lambs spent grazing resulted in similar loin IMF content as the 3-weeks milk-fed lambs. This result contrasts with the higher IMF content in meat observed at older lamb ages in other studies that did not wean the lambs [19] or fed them a high-concentrate diet after weaning rather than pasture [20,21]. In addition, the different results observed in the present study do not seem to be associated with the animal genetic background (i.e., dairy vs. meat breeds) as dairy breeds were also used in the mentioned studies by Beriain, Horcada, Purroy, Lizaso, Chasco, and Mendizabal [20] and Martínez-Cerezo, Sañudo, Panea, Medel, Delfa, Sierra, Beltrán, Cepero, and Olleta [21]. The treatment effects on loin IMF content are especially important, as it can affect not only the meat fatty acid profile [22] but also its sensory characteristics [4].

The IMF content obtained in the present study (approx. 2.8%) was similar to the average of 2.7% previously reported for New Zealand lambs from meat breeds slaughtered between 3 and 8 months old [4,23]. It should be highlighted that these previous studies used composite genetics of lambs specialized in meat production, whereas the lambs in the present study were dairy genetics (East Friesian × Lacaune crossbred lambs). Therefore, the results of this study suggest that the IMF content of dairy lambs reared either naturally or artificially and finished in outdoor pasture farming systems in the New Zealand temperate climate can produce lamb meat with similar IMF content to lambs from meat genetic backgrounds.

### 4.2. Rearing System and Slaughter Age Effects on Meat Fatty Acid Composition

Meat TVA and CLA originate externally from animals’ diet or internally from ruminal bacterial production [24,25]. As milk-fed lambs with <3 weeks of age do not have a fully functional rumen [26], these fatty acids in their meat are mainly from the ingested milk [27]. Hence, the higher content observed of those fatty acids in meat from the NR compared to the AR lambs would be associated with the type and/or volume of milk ingested. In agreement with this, sheep’s milk has been reported to contain greater CLA proportions than cow’s milk [28,29]. Though, the higher TVA proportions in meat from NR than AR lambs contrast with the lower proportions of this fatty acid reported in ewe’s than in cows’ milk from New Zealand pasture-fed outdoor dairy production systems [29]. Nonetheless, the milk replacer used in the present study had a lower TVA proportion than that reported by Teng, Reis, Yang, Ma, and Day [29] for the cows’ milk (2.9% vs. 4.1%). Similarly, the lower C14:1 proportion observed in NR meat would be associated with the different proportions of this fatty acid in their dietary milk. Ovine milk has been suggested to contain a lower proportion of C14:1 than cow milk [28,29]. However, it needs to be considered that, in the present study, milk intake could not be measured. And therefore, the potential contribution of milk intake to the differences observed in the FA profile of meat cannot be determined.

In contrast to the low number of fatty acids and ratio differences between NR and AR observed in the present study, many more differences were observed in other studies [27,30,31,32], where vegetable oil rather than milk fat was used in the milk replacer. One important difference between both types of dietary fat sources is their proportion of PUFAs. The milk replacer used in the present study had 2.6% of total PUFAs, whereas the milk replacer containing 100% vegetable oil reported in prior studies had higher PUFA content; 42.6% [30], 9.7% [31], and 7.2% [27]. Therefore, our results suggest that the artificial rearing of lambs on a bovine milk-based milk replacer or naturally on ovine whole milk has a minor influence on lamb meat fatty acid profile.

The greater desaturation ratios (C16:1/C18:0, C18:1/C18:0, and CLA/TVA) observed in the meat from the 3-weeks milk-fed than from the 3-months pasture-fed lambs would be a reflection of their dietary fatty acid composition since at that age the rumen would still be developing [33]. As weaned lambs were grazed on pasture, the ratios would have been reduced in the 3-months pasture-fed lambs by a low activity of the stearoyl-CoA desaturase in these types of diets [34]. Furthermore, in the present study, as was suggested by Bas and Morand-Fehr [33], meat proportions of C14:0 and C14:1 decreased after weaning, but the proportions of the longer chain fatty acids, C18:0 and C18:1, increased.

In agreement with the present study, others [19,20,35] also observed that increasing slaughter age reduces PUFAs proportions. However, in contrast to Beriain, Horcada, Purroy, Lizaso, Chasco, and Mendizabal [20], Camacho, Torres, Capote, Mata, Viera, Bermejo, and Argüello [35], and della Malva, Albenzio, Annicchiarico, Caroprese, Muscio, Santillo, and Marino [19], the reduction of PUFA proportion observed in the present study was not associated with an increase of total fatty acid content with slaughter age. The 3-weeks milk-fed lambs could have had a greater PUFA proportion as these fatty acids would have not been hydrogenated in the rumen due to its poor fermentation capacity [33]. On the other hand, although in the present study, the SFA and MUFA proportions increased with slaughter age, others observed that either the SFA or MUFA proportion increased while the other decreased [19,20,35].

### 4.3. Rearing System and Slaughter Age Effects on Consumer Liking of Lamb Meat

In general, no rearing system effects were observed on the liking scores from all consumers, but a trend of a rearing system difference between clusters was observed for the overall liking scores, most likely driven by the trends observed in flavor and juiciness liking. This small change of flavor liking between clusters when comparing the rearing systems does not appear to be associated with changes in fatty acid proportions between the rearing systems.

When looking at the slaughter age effect on liking scores from all consumers, it was observed that in general, consumers had higher overall liking scores for meat from 3-weeks milk-fed than meat from 3-months pasture-fed lambs. This preference was associated with the higher tenderness and juiciness liking scores but was not associated with any difference in flavor liking scores.

The meat tenderness preference for meat from younger lambs by all consumers in this study agrees with the consumer preference reported by della Malva, Albenzio, Annicchiarico, Caroprese, Muscio, Santillo, and Marino [19] for meat from 45 than 70 d old milk-fed lambs, but contrast with the similar consumer scores obtained by Sañudo et al. [36] when meat from 57, 72, or 97 days old weaned (40–50 days) lambs were compared. Sañudo, Santolaria, María, Osorio, and Sierra [36], in contrast to the present study, fed a concentrate rather than a pasture diet after weaning. On the other hand, the higher liking scores for meat juiciness from 3-weeks milk-fed lambs in the present study contrast with the higher scores reported for meat from older lambs in both studies [19,36] which could be attributed to their higher IMF content [6]. Finally, as in the present study, neither della Malva, Albenzio, Annicchiarico, Caroprese, Muscio, Santillo, and Marino [19], nor Sañudo, Santolaria, María, Osorio, and Sierra [36] observed a slaughter age or slaughter weight effect on consumers’ meat flavor scores. In the present study, it was observed that consumers from Cluster-1 gave the same importance to all three sensory attributes in the definition of overall liking, whereas consumers from Cluster-2 defined the overall liking based on their flavor liking scores.

In general, it has been observed that consumers like the flavor of the meat they are used to [11,12]. Sañudo, Alfonso, San Julián, Thorkelsson, Valdimarsdottir, Zygoyiannis, Stamataris, Piasentier, Mills, Berge, Dransfield, Nute, Enser, and Fisher [11] found two main types of consumers of lamb meat in Europe: those that preferred the taste of milk- or concentrate-fed lambs and those that preferred the taste of grass-fed lambs based on the production system in their region. According to these authors, meat from milk- or concentrate-fed lambs has a paler and more delicate flavor that contrasts with meat from pasture-fed lambs, which has a fatter and more robust flavor. It is interesting to note that, although the current study was performed with New Zealand consumers, which are used to meat from pasture-fed animals (beef and lambs), a greater proportion (60%) preferred the meat from the 3-weeks milk-fed over 3-months pasture-fed lambs. These results suggest that, just like the European consumers from the Mediterranean region, a greater proportion of New Zealand consumers preferred the delicate lamb flavor of the milk-fed lambs.

### 4.4. Relationship between Consumer Liking Scores and IMF and Fatty Acid Composition of Lamb

Intramuscular fat has been reported to have a positive effect on eating quality, as increasing levels in meat are associated with increasing tenderness, flavor, and juiciness [4,6]. Significant positive correlations between IMF and flavor and overall liking scores were obtained in this study for consumers in Cluster-2. but correlations were not significant in Cluster-1. Previous studies have shown that not only the content but also the composition of IMF influence consumer liking scores [4,12,19,37]. SFA and MUFA proportions in this study were positively and PUFA proportions were negatively associated with flavor liking scores for consumers in Cluster-2 only. It seems that the liking of flavor by consumers in Cluster-1 was mainly driven by non-lipidic metabolites [38]. This would be in agreement with the different sensitivity for fatty acids among consumers suggested by Stewart et al. [39].

Heptadecanoic acid (C17:0) has been proposed as a marker for volatile BCFA and linked with mutton-like flavor [40]. In agreement with other authors [41,42], the meat proportion of C17:0 increased with animal age. Therefore, the negative correlations between C17:0 and flavor or overall liking of meat in Cluster-1 but positive in Cluster-2, indicate a divergence in consumer preferences between clusters, with consumers in Cluster-1 preferring a milder sheep meat flavor than those in Cluster-2. Similar divergence in preferences for lamb meat between consumers was reported by Pavan, Ye, Eyres, Guerrero, Reis, Silcock, Johnson, and Realini [37]. The fact that the flavor liking from consumers in Cluster-1 was also negatively correlated with the PUFA C18:3 *n*-3, which is associated with the flavor of forage-fed animals [43], also suggests that these consumers would prefer lamb with a mild pasture flavor. On the other hand, as also observed by della Malva, Albenzio, Annicchiarico, Caroprese, Muscio, Santillo, and Marino [19], scores from consumers in Cluster-1 showed a strong negative correlation (r = −0.80) between tenderness and meat C18:0 proportion. As the proportion of C18:0 decreases, the fat melting point decreases, and tenderness increases [44].

## 5. Conclusions

The rearing systems used in the present study did not affect meat sensory properties, but the slaughter age of lambs influenced consumer liking scores. Most consumers preferred on average meat from 3-weeks milk-fed lambs than meat from 3-months pasture-fed ones. In agreement with this, the slaughter age of lambs had a stronger influence on the fatty acid composition of meat than the rearing system, which only influenced the proportions of CLA and TVA. Two groups/clusters of lamb consumers emerged from the present study, with one cluster (60% of consumers) preferring meat from 3-weeks milk-fed lambs compared to 3-months pasture-fed lambs. The preference for meat from 3-weeks lambs by Cluster-1 was driven by all sensory attributes but mainly by the much higher tenderness liking scores than meat from 3-months lambs. The overall liking for this cluster was not correlated with IMF, but some correlations were evident between tenderness and flavor liking with few fatty acids. On the other hand, the higher overall liking of meat from 3-months lambs by Cluster-2 was driven by flavor liking only, indicating a preference for stronger lamb flavor from older lambs finished on pasture. The overall liking for this cluster was positively correlated with IMF, and several correlations were significant between flavor and many total and individual meat fatty acids.

The outcomes of the present study support a niche market opportunity for the utilization of surplus lambs, which are a by-product of the dairy sheep industry, due to the higher preference by consumers of meat from 3-week-old lambs compared to meat from older lambs.

## Figures and Tables

**Table 1 foods-11-02350-t001:** Socio-demographic characteristics for all consumers and their clusters.

Characteristics, % of Total	All Consumers	Cluster-1	Cluster-2	Cluster Effect
				*p*-Value Chi-Square
*n*	132	79	53	
Age				
20–29 years	32.5	24.7	44.0	0.1279
30–39 years	28.5	34.2	20.0	
40–49 years	17.9	16.4	20.0	
50–59 years	17.9	20.6	14.0	
>60 years	3.2	4.1	2.0	
Gender				
Female	62.9	63.3	62.3	0.9781
Male	37.1	36.7	37.7	
Lamb consumption frequency				
Daily	0.8	1.3	0.0	0.4917
2–3 times a week	13.8	10.3	19.2	
Once per week	34.6	39.7	26.9	
Once per fortnight	26.9	25.6	28.8	
Once per month	18.5	18.0	19.2	
Less frequently	5.4	5.1	5.8	

**Table 2 foods-11-02350-t002:** Effect of rearing system and slaughter age on m. longissimus intramuscular fat (IMF) content (% fresh tissue) and fatty acid composition (12 animals per treatment; animals used in the sensory panel).

Slaughter Age, SA	3 Weeks		3 Months		*p*-Value			Pearson’s Correlation Coefficients α with IMF
Rearing System, RS	Natural (*n* = 2)	Artificial (*n* = 4)	Natural (*n* = 3)	Artificial (*n* = 4)	RS	SA	RS × SA	
Intramuscular fat, % fresh tissue	2.74 ± 0.22	2.76 ± 0.22	2.77 ± 0.19	2.77 ± 0.19	0.9664	0.9265	0.9758	
Fatty acid composition, % of total fatty acid content								
Total SFAs ^1^	36.63 ± 0.91	38.56 ± 0.85	40.85 ± 0.88	40.77 ± 0.85	0.3935	0.0002	0.2095	0.59 ***
C10:0 ^δ^	0.15 ± 0.03	0.08 ± 0.03	0.10 ± 0.02	0.07 ± 0.02	0.0736	0.1328	0.1918	0.51 ***
C12:0	0.66 ± 0.05	0.58 ± 0.05	0.54 ± 0.05	0.57 ± 0.05	0.6081	0.1677	0.3004	−0.24 ^t^
C14:0 ^δ^	3.57 ± 0.31	4.13 ± 0.31	1.65 ± 0.16	2.11 ± 0.16	0.0784	<0.0001	0.8400	0.40 **
C15:0	0.36 ± 0.03	0.37 ± 0.03	0.34 ± 0.03	0.40 ± 0.03	0.3577	0.7809	0.4130	0.37 **
C16:0	17.45 ± 0.82	20.00 ± 0.78	19.29 ± 0.80	19.26 ± 0.78	0.2179	0.4617	0.0911	0.41 **
C17:0	0.75 ± 0.04	0.66 ± 0.04	0.94 ± 0.05	0.96 ± 0.05	0.4608	<0.0001	0.2322	0.34 *
C18:0	13.77 ± 0.44	12.74 ± 0.43	17.98 ± 0.45	17.39 ± 0.44	0.1507	<0.0001	0.6376	0.05
Total BCFAs ^2^	0.95 ± 0.05	0.87 ± 0.05	0.96 ± 0.06	0.99 ± 0.05	0.7403	0.1543	0.2690	0.31 *
Iso C15:0	0.09 ± 0.02	0.09 ± 0.01	0.12 ± 0.02	0.12 ± 0.01	0.8466	0.0524	0.9773	0.61 ***
Anteiso C15:0	0.15 ± 0.01	0.16 ± 0.01	0.19 ± 0.01	0.20 ± 0.01	0.3992	0.0034	0.6397	0.05
Iso C16:0	0.13 ± 0.01	0.12 ± 0.01	0.11 ± 0.01	0.13 ± 0.01	0.8988	0.8116	0.3833	0.43 **
Iso C17:0	0.58 ± 0.03	0.50 ± 0.03	0.53 ± 0.03	0.54 ± 0.03	0.2721	0.8990	0.1764	0.04
Total MUFAs ^3^	37.65 ± 1.16	38.02 ± 1.03	40.59 ± 1.09	39.74 ± 1.03	0.8810	0.0141	0.4833	0.46 ***
C14:1 ^δ^	0.16 ± 0.05	0.37 ± 0.05	0.00 ± 0.02	0.15 ± 0.02	<0.0001	<0.0001	0.4426	0.13
C16:1 ^δ^	1.27 ± 0.33	2.10 ± 0.33	0.96 ± 0.22	1.71 ± 0.20	0.0568	0.0687	0.9183	0.08
C17:1	0.57 ± 0.03	0.55 ± 0.03	0.76 ± 0.03	0.73 ± 0.03	0.3487	<0.0001	0.9645	−0.13
C18:1 *trans*-9	0.20 ± 0.01	0.18 ± 0.01	0.21 ± 0.01	0.19 ± 0.01	0.0901	0.5397	0.8147	0.58 ***
C18:1 *trans*-11, TVA ^δ^	2.98 ± 0.23	1.99 ± 0.23	3.30 ± 0.32	2.87 ± 0.32	0.0148	0.0365	0.3117	0.31 *
C18:1 *cis*-9	30.69 ± 0.96	31.14 ± 0.87	34.28 ± 0.91	32.83 ± 0.88	0.6530	0.0015	0.2301	0.41 **
C18:1 *cis*-11 ^δ^	1.57 ± 0.09	1.69 ± 0.09	1.09 ± 0.06	1.26 ± 0.06	0.1198	<0.0001	0.7570	−0.55 ***
Total PUFAs ^4^	18.82 ± 1.56	16.45 ± 1.42	11.76 ± 1.48	12.96 ± 1.42	0.7765	0.0002	0.1822	−0.60 ***
C18:2 *n*-6 ^δ^	6.58 ± 0.64	5.72 ± 0.64	3.99 ± 0.34	4.41 ± 0.34	0.6660	0.0005	0.2181	−0.49 ***
C18:3 *n*-3 ^δ^	2.18 ± 0.14	1.90 ± 0.14	2.18 ± 0.24	2.13 ± 0.24	0.5097	0.5860	0.5697	−0.53 ***
C18:2 *cis*-9 *trans*-11, CLA	1.51 ± 0.07	1.23 ± 0.07	1.06 ± 0.07	0.94 ± 0.07	0.0062	<0.0001	0.2856	0.27 ^t^
C20:4 *n*-6	3.07 ± 0.42	2.60 ± 0.38	1.77 ± 0.40	2.25 ± 0.38	0.9905	0.0120	0.1420	−0.59 ***
C20:5 *n*-3	1.97 ± 0.24	1.72 ± 0.21	1.19 ± 0.22	1.36 ± 0.21	0.9119	0.0021	0.2574	−0.68 ***
C22:5 *n*-3	2.25 ± 0.24	2.16 ± 0.21	1.21 ± 0.23	1.51 ± 0.21	0.6814	<0.0001	0.3022	−0.66 ***
C22:6 *n*-3 ^δ^	1.25 ± 0.11	1.10 ± 0.11	0.37 ± 0.04	0.37 ± 0.04	0.4205	<0.0001	0.3879	−0.36 *
Total *n*-6 PUFA ^δ,5^	9.57 ± 1.00	8.32 ± 0.95	5.76 ± 0.75	6.66 ± 0.72	0.8593	0.0011	0.1736	−0.55 ***
Total *n*-3 PUFA ^6^	7.59 ± 0.66	6.89 ± 0.61	4.94 ± 0.63	5.36 ± 0.61	0.8525	0.0006	0.3215	−0.68 ***
Ratios								
PUFAs/SFAs	0.49 ± 0.05	0.46 ± 0.05	0.28 ± 0.03	0.31 ± 0.03	0.9556	<0.0001	0.4219	
*n*-6/*n*-3	1.26 ± 0.07	1.20 ± 0.06	1.19 ± 0.07	1.28 ± 0.06	0.8339	0.9039	0.2337	0.14
C16:1/C18:0 ^δ^	0.11 ± 0.03	0.17 ± 0.03	0.06 ± 0.01	0.10 ± 0.01	0.0324	0.0016	0.5142	0.03
C16:1/C16:0 ^δ^	0.08 ± 0.016	0.10 ± 0.015	0.05 ± 0.011	0.09 ± 0.010	0.0864	0.0721	0.4638	−0.03
C18:1/C18:0 ^δ^	2.27 ± 0.14	2.49 ± 0.13	1.92 ± 0.10	1.90 ± 0.09	0.5674	<0.0001	0.2663	0.12
CLA/TVA ^δ^	0.54 ± 0.036	0.65 ± 0.036	0.33 ± 0.026	0.36 ± 0.026	0.0373	<0.0001	0.2273	−0.27

^δ^ heterogeneous variance between slaughter age was included. ^t^, *, ** and *** refer to *p* < 0.10, *p* < 0.05, *p* < 0.01, and *p* < 0.001, respectively. ^1^ Total saturated fatty acids (SFAs) = Σ C10:0, C12:0, C14:0, C15:0, C16:0, C17:0, C18:0. ^2^ Total branched-chain fatty acids (BCFAs) = Σ isoC15:0, anteiso-C15:0, isoC16:0, iso-C17:0. ^3^ Total MUFAs (monounsaturated fatty acids) = Σ C14:1, C16:1, C18:1 trans-9, C18:1 trans-11, C18:1 cis-9, C18:1 cis-11. ^4^ Total polyunsaturated fatty acids (PUFAs) = Σ C18:2 *n*-6, C20:4 *n*-6, C18:3 *n*-3, C20:5 *n*-3, C22:5 *n*-3, C22:6 *n*-3, C18:2 cis-9, trans-11. ^5^ Total *n*-6 PUFA = Σ C18:2 *n*-6, C20:4 *n*-6. ^6^ Total *n*-3 PUFA = Σ C18:3 *n*-3, C20:5 *n*-3, C22:5 *n*-3, C22:6 *n*-3.

**Table 3 foods-11-02350-t003:** Effect of lamb rearing system and slaughter age on consumer liking of meat (Lsmeans ± SEM).

Parameter ^1^	Rearing System		Slaughter Age		*p*-Values			
	Natural	Artificial	3-Weeks	3-Months	RS	SA	RS × SA	IMF ^2^ (Covariate)
Liking of tenderness	69.11 ± 1.61	66.86 ± 1.61	78.13 ± 1.67	57.84 ± 1.55	0.2331	<0.0001	0.2221	0.0002
Liking of juiciness	61.17 ± 1.69	60.65 ± 1.69	69.05 ± 1.74	52.76 ± 1.65	0.7599	<0.0001	0.2133	0.0426
Flavor liking	63.58 ± 1.52	62.13 ± 1.52	62.90 ± 1.62	62.82 ± 1.41	0.3669	0.9583	0.9035	0.0001
Overall liking	65.13 ± 1.53	63.69 ± 1.53	68.68 ± 1.64	60.14 ± 1.41	0.3993	<0.0001	0.7828	0.0001

^1^ Consumer liking scores from 0: dislike extremely to 100: like extremely. IMF ^2^: intramuscular fat.

**Table 4 foods-11-02350-t004:** Consumer clusters (cluster), lamb rearing system, and slaughter age effects on consumer liking of meat.

	Consumer Liking ^1^			
	Tenderness	Juiciness	Flavor	Overall
*p*-values				
Rearing system, RS	0.2533	0.9848	0.5284	0.5532
Slaughter age, SA	<0.0001	<0.0001	0.0624	0.0006
RS × SA	0.1613	0.1925	0.8246	0.6689
Cluster	<0.0001	0.0063	<0.0001	<0.0001
Cluster × RS	0.3493	0.0639	0.0604	0.0126
Cluster × SA	<0.0001	<0.0001	<0.0001	<0.0001
Cluster × RS × SA	0.6906	0.7371	0.6688	0.6957
IMF (covariate)	0.0012	0.0787	0.0001	0.0001
**Consumer liking scores (lsmeans ± SEM)**				
**Cluster** × **Rearing system**				
Cluster-1: Natural	74.1 ± 1.9	65.3 ± 2.1	70.2 ± 1.8	71.6 ± 1.7 ^A^
Artificial	70.7 ± 1.9	62.8 ± 2.1	66.5 ± 1.8	67.6 ± 1.7 ^B^
Cluster-2: Natural	61.6 ± 2.4	55.0 ± 2.5	53.7 ± 2.2	55.4 ± 2.1 ^C^
Artificial	61.1 ± 2.3	57.5 ± 2.5	55.6 ± 2.2	57.9 ± 2.1 ^C^
**Cluster × Slaughter age**				
Cluster-1: 3-weeks	88.8 ± 1.9 ^a^	77.4 ± 2.1 ^a^	74.0 ± 1.8 ^a^	80.0 ± 1.7 ^a^
3-months	56.0 ± 2.0 ^b^	50.7 ± 2.0 ^b^	62.7 ± 1.8 ^b^	58.2 ± 1.7 ^b^
Cluster-2: 3-weeks	62.2 ± 2.3 ^b^	56.6 ± 2.5 ^b^	46.3 ± 2.3 ^c^	50.3 ± 2.1 ^c^
3-months	60.6 ± 2.4 ^b^	55.9 ± 2.5 ^b^	63.0 ± 2.2 ^b^	63.1 ± 2.1 ^b^

^A–C^ Different superscript letters within columns for cluster by rearing, or for cluster by slaughter age, denote a tendency (*p* ≤ 0.10) toward a significant difference between values according to Tukey’s test. ^a–c^ Different superscript letters within columns for cluster by rearing, or for cluster by slaughter age, denote significant differences (*p* ≤ 0.05) between values according to Tukey’s test. ^1^ Consumer liking scores from 0: dislike extremely to 100: like extremely.

## Data Availability

The data presented in this study are available on request from the corresponding author. Although consumer data have been anonymized, data are not publicly available.

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
