# Peer review of "Effects of Dairy Lambs’ Rearing System and Slaughter Age on Consumer Liking of Lamb Meat and Its Association with Lipid Content and Composition"

_foods, 2022, doi:10.3390/foods11152350_

Round 1

Reviewer 1 Report

Please see comments given in the reviewed attached file of manuscript.

Author Response

Reviewer 1:

  1. Add some sentences about materials and methods in the abstract.

There is not enough space to expand the abstract. It is mentioned in the abstract that “consumer liking of dairy lamb Longissimus lumborum muscle and its association with lipid content and composition were evaluated

  1. Add conclusions at the end of the abstract.

A brief conclusion is presented at the end of the abstract: “Meat fatty acids profile and consumer liking were not influenced by the rearing system but by lamb slaughter age; showing a niche product opportunity for the 3-wk milk‑fed lambs”. Due to the word limit of 200 words it cannot be further expanded.

  1. It is better to explain about importance of meat production and meat quality in farm animals. For this, you can sue below added sentences and references:

Moreover, farm-animal species play crucial roles in satisfying demands for meat on a global scale, and they are genetically being developed to enhance the efficiency of meat production (Amiri Roudbar et al., 2017; Masoudzadeh et al., 2020). In particular, one of the important breeders’ aims is to increase skeletal muscle growth in farm animals (Nassiry et al., 2005; Zamani et al., 2015; Arabpour et al., 2021). The enhancement of muscle development and growth is crucial to meet consumers’ demands regarding meat quality (Mohammadi et al., 2009; Mohammadabadi et al., 2021).

  • Amiri Roudbar M, Mohammadabadi MR, Mehrgardi AA, Abdollahi-Arpanahi A 2017. Estimates of variance components due to parent-of-origin effects for body weight in Iran-Black sheep. Small Ruminant Research 149, 1-5.
  • Arabpour Z, Mohammadabadi M, Khezri A 2021. The expression pattern of p32 gene in femur, humeral muscle, back muscle and back fat tissues of Kermani lambs. Agricultural Biotechnology Journal 13 (4), 183-200.
  • Masoudzadeh, S.H., Mohammadabadi, M., Khezri, A., Stavetska, R.V., Oleshko, V.P., Babenko, O.I., Yemets, Z., Kalash

We do not consider that further explanation of the importance of meat production and meat quality is necessary for the manuscript. The focus is on the effect the rearing system and the slaughter age have on consumer liking and its association with lipid content and composition and that is what is intended to be described in the introduction.

  1. please add used animal model and uts components in the text of manuscript.

Change made. The initial paragraph “The relationships between consumer panel scores from both clusters, and between consumer panel scores and the meat fatty acid profile were assessed using the correlation (CORR) procedure in SAS.” was replaced by “The relationships between the average consumer panel scores per lamb from both clusters, and between the average consumer panel scores and the meat fatty acid profile per lamb were assessed using the correlation (CORR) procedure in SAS (n = 48).”

Reviewer 2 Report

Effects of dairy lambs’ rearing system and slaughter age on consumer liking of lamb meat and its association with lipid content and composition

General comments:

The authors studied the effects of dairy lamb’s rearing system and slaughter age on consumer acceptability of lamb meat and their interaction on FFA. Overall, the obtained results are well presented. The experimental design was well described, however the demographics of the participant is not well elaborated (e.g. ethnic origin, etc.). Pre survey question on the frequency and percentage of participants response towards lamb is also not included (e.g. frequency of eating lamb, method of cooking, degree of doneness) to know the acceptance/preference of consumers. Maybe some judgement will be error due to the consumer biased towards the off-odor or taste of lamb.  The result and discussion are presented in an informative way, and the authors explain the results and discussion very well. Only some parts in this manuscript that needs to be addressed:

Introduction

Overall introduction is very well written but cited references used are not the new/updated version (1988-2015). Suggest replace/adding some reference with the latest one.

Materials and methods

L88- remove)

Results and discussions

Table 1 – authors mentioned the number of respondents in L130 was 132 but in Table 1 133 consumers

The results and discussions are well constructed and argued with many supported references. Reviewer has no further comment on this part.

Author Response

Reviewer 2:

  1. General comments:

The authors studied the effects of dairy lamb’s rearing system and slaughter age on consumer acceptability of lamb meat and their interaction on FFA. Overall, the obtained results are well presented. The experimental design was well described, however the demographics of the participant is not well elaborated (e.g. ethnic origin, etc.). Pre survey question on the frequency and percentage of participants response towards lamb is also not included (e.g. frequency of eating lamb, method of cooking, degree of doneness) to know the acceptance/preference of consumers. Maybe some judgement will be error due to the consumer biased towards the off-odor or taste of lamb. The result and discussion are presented in an informative way, and the authors explain the results and discussion very well. Only some parts in this manuscript that needs to be addressed:

All the demographics of the participant obtained (age, gender and lamb consumption frequency) are presented in Table 1. We agree that more information from the participant (ethnic origin, method of cooking, degree of doneness, etc) would have been valuable, but is not available.

  1. Introduction

Overall introduction is very well written but cited references used are not the new/updated version (1988-2015). Suggest replace/adding some reference with the latest one.

More recent citations had been added in the introduction.

  • Realini, C.E.; Pavan, E.; Johnson, P.L.; Font, I.F.M.; Jacob, N.; Agnew, M.; Craigie, C.R.; Moon, C.D. Consumer liking of M. longissimus lumborum from New Zealand pasture-finished lamb is influenced by intramuscular fat. Meat Sci. 2021, 173, 108380, doi:10.1016/j.meatsci.2020.108380.
  • Phelps, M.R.; Garmyn, A.J.; Brooks, J.C.; Mafi, G.G.; Duckett, S.K.; Legako, J.F.; O’Quinn, T.G.; Miller, M.F. Effects of marbling and postmortem aging on consumer assessment of United States lamb loin. Meat Muscle Biol. 2018, 2, 221-232, doi:10.22175/mmb2017.09.0045.
  • Lambe, N.R.; McLean, K.A.; Gordon, J.; Evans, D.; Clelland, N.; Bunger, L. Prediction of intramuscular fat content using CT scanning of packaged lamb cuts and relationships with meat eating quality. Meat Sci. 2017, 123, 112-119, doi:https://doi.org/10.1016/j.meatsci.2016.09.008.
  • Battacone, G.; Lunesu, M.F.; Rassu, S.P.G.; Pulina, G.; Nudda, A. Effect of Dams and Suckling Lamb Feeding Systems on the Fatty Acid Composition of Suckling Lamb Meat. Animals (Basel) 2021, 11, doi:10.3390/ani11113142.
  1. Materials and methods

L88- remove).

Do not understand what the reviewer means. We consider that it is important to mention that “fresh water was available ad libitum for all animals.”

  1. Results and discussions

Table 1 – authors mentioned the number of respondents in L130 was 132 but in Table 1 133 consumers.

The total number of consumers was 132, Table 1 was corrected.

The results and discussions are well constructed and argued with many supported references. Reviewer has no further comment on this part.

Reviewer 3 Report

The research work is well-conceived and well-executed. I think the manuscript can be further improved after minor spell/grammar check.

Author Response

Reviewer 3

The research work is well-conceived and well-executed. I think the manuscript can be further improved after minor spell/grammar check.

Thank you for your positive feedback

Reviewer 4 Report

The manuscript evaluates the effects of lamb rearing system and age at slaughter on consumer preferences for lamb meat, and their relationship to the fatty acid content. The subject of this study is of interest both to the readers of the journal and to the lamb meat and dairy industry.

The experimental approach is very well thought out and, although the results do not indicate a significant effect of some of the factors considered, my opinion is that the information is scientifically relevant and very interesting for the lamb industry. As the authors indicate in their conclusions, I agree that the results of the study support a niche market for the utilization of surplus lambs that are a by-product of the dairy sheep industry.

The manuscript is well written, clear and structured. The methods used are appropriate and well described. The experiment is well designed and executed. The introduction, results and discussion sections are well supported by adequate references.  

Author Response

Thank you for your positive feedback

Reviewer 4

The manuscript evaluates the effects of lamb rearing system and age at slaughter on consumer preferences for lamb meat, and their relationship to the fatty acid content. The subject of this study is of interest both to the readers of the journal and to the lamb meat and dairy industry.

The experimental approach is very well thought out and, although the results do not indicate a significant effect of some of the factors considered, my opinion is that the information is scientifically relevant and very interesting for the lamb industry. As the authors indicate in their conclusions, I agree that the results of the study support a niche market for the utilization of surplus lambs that are a by-product of the dairy sheep industry.

The manuscript is well written, clear and structured. The methods used are appropriate and well described. The experiment is well designed and executed. The introduction, results and discussion sections are well supported by adequate references.  

Thank you for your positive feedback.